Application of material from used car tyres in geotechnics—an environmental impact analysis

Duda Aleksander aduda@prz.edu.pl 1
Kida Małgorzata 2
Ziembowicz Sabina 2
Koszelnik Piotr 2
1 Department of Roads and Bridges, Faculty of Civil and Environmental Engineering and Architecture, Rzeszow University of Technology , Rzeszow , Poland
2 Department of Chemistry and Environmental Engineering, Faculty of Civil and Environmental Engineering and Architecture, Rzeszow University of Technology , Rzeszow , Poland
Swaminathan Meenakshisundaram
Electronic publication date: 2020 Jul 20
Publication date: 2020
Volume: 8
Electronic Location ID: e9546
Received 2020 Feb 13; Accepted 2020 Jun 24
Copyright: ©2020 Duda et al.
Copyright year: 2020
Copyright holder: Duda et al.
License: This is an open access article distributed under the terms of the Creative Commons Attribution License, which permits unrestricted use, distribution, reproduction and adaptation in any medium and for any purpose provided that it is properly attributed. For attribution, the original author(s), title, publication source (PeerJ) and either DOI or URL of the article must be cited.
License URL: https://creativecommons.org/licenses/by/4.0/

Keywords: Tyre recycling, Tyre bales, Rainwater, Groundwater, Environment, Pollutant leaching, Building material, Research project

Funding: “ReUse—Innovative Recycling Materials, Enhancing the Sustainability of Bridge Facilities” K3/IN3/38/228116/NCBiR/15 European Regional Development Fund This research has been carried out under the project called “ReUse—Innovative Recycling Materials, Enhancing the Sustainability of Bridge Facilities” (Innotech No. K3/IN3/38/228116/NCBiR/15), co-financed by the European Regional Development Fund and implemented by the consortium of Promost Consulting Rzeszow (leader of consortium), Remost Debica, Geotech Rzeszow and Rzeszow University of Technology. The funders had no role in study design, data collection and analysis, decision to publish, or preparation of the manuscript.

==============================
This work begins with a literature-based discussion of the hazardous-waste problem represented by car tyres as hazardous waste, along with possible ways in which they might be utilised or managed. The impact of the material on the environment is characterised in the process, not least in the context of pollutants leached to the aquatic environment. Input in terms of new research results concerns the impact on water and soil of material from used car tyres being used in geotechnics. Specifically, tyre bales comprising 100–140 car vehicle tyres compressed into a lightweight block and secured by galvanised steel tie wires running around the length and depth of the bale, were researched, having been immersed in basins with alkaline and acidic water following initial preparation and pre-washing. The aim was to in some sense simulate—respectively—conditions in which rain and surface/ground water are involved, or else acid rain. To do that, the tyre bales were placed in the water for 120 days, with emerging leachate analysed after set intervals of time, with a view to changes in key physicochemical parameters of water being noted, as well as signs of the leaching of both undesirable components and priority substances, from tyres into the aqueous medium. Washing of the tyre bales was shown to induce slight pollution of water, with limited exceedance of normative values in respect of OWO content. However, this increase was not due to leaching of the Persistent Organic Pollutants tested for, but may rather have reflected contamination of tyres used, e.g., of soil at the place of previous storage. In general, waste water arising does not therefore contain substances that would stand in the way (legally) of its being discharged into a combined sewer system. Similar conclusions were arrived at through analysis of the leaching of pollutants from tyre bales exposed in the aforementioned pools of water of neutral and acidic reaction. Wastewater arising was not enriched significantly in impurities (be these metals, PAHs, phthalates, selected anions or cations), and there were therefore no exceedances of standards imposed for wastewater discharged to either waters or soil.

Introduction

Globally, the rubber and car industries generates hundreds of millions of used tyres each year. That mass is foreseen to increase steadily, at a rate in line with the increase in the number of motor vehicles. In the EU as a whole, 2017 saw around 3.4 million tons of used tyres of passenger cars generated, while figures for Poland include 275,000 tonnes (along with 200,000 tonnes of truck tyres) (Central Statistical Office, 2017). Similar data on the production of passenger car tyres in Poland are available from the European tyre and Rubber Manufacturers Association—ETRMA. The conclusion (ETRMA, 2017) would thus be that 79% of the tyres produced in Poland are recycled or recovered, with 42% becoming recycled material (granulation tyres) or retread, along with 29% going for energy recovery (burning in cement kilns) or energy recycling (pyrolysis). The fact that the remaining 21% of spent tyres go unused must represent great potential for the tyres to gain use in other parts of the economy, e.g., in civil engineering. Rubber material from tyre-recycling (in fine or granulated form) does gain successful use in road construction, where the application is to modify the composition of asphalt (Liu, Cai & Liu, 2018). The bituminous surfaces arising from this activity are demonstrably more durable and more flexible, of lesser luminance, and of greater resistance to abrasion. They also achieve greater roughness, while serving to reduce the level of noise arising as tyres still in use on vehicles make contact.

In addition, rubber also gains use in the manufacture of clothing, car mats, mats for farm animals, wipers, playground surfaces, and so on (Duda et al., 2016). However, at present there are no examples in Poland of such pro-ecological solutions as direct tyre-recycling processes, whereby waste material processed to only a limited extent generates products of new and desired features. In countries such as the USA, France, Japan, the UK and Spain, as well as the Scandinavian countries, rubber aggregate in the form of shreds and chips is most often used as filler in earth and engineering constructions (embankments, retaining structures, backfill of abutments or drainage infrastructure). Similar applications appear where the combined form of the tyre bale is made use of. The first key advantage of such limited either fragmentation or compression of tyres is the way in which the recycling process then only takes between one-tenth or one-sixteenth as much energy as does the fragmentation of granular tyres, or tyre pyrolysis (Winter, Watts & Johnson, 2006).

The high level of energy consumption inherent in far-reaching recycling of used tyres reflects the complex construction, wherein rubber is augmented by both textile and steel cord. This leaves used car tyres as extremely durable waste not naturally inclined to decompose and resistant to the action of water, various chemicals and extreme temperatures alike. This in turn means a major long-term threat associated with the possibility of fire taking hold at dumps where large numbers of tyres are present, as a result of vandalism (arson), the self-ignition of loose tyres or the role of nature (as with a lightning strike). Table 1 reports on selected cases of fire involving tyres that were being stored, in which the level of environmental hazard is very high indeed, for tyre fires release harmful chemicals capable of polluting air, water and soil.

Table 1 Selected landfill tyre fires (HPA & Chemical Hazards and Poisons Report From the Chemical Hazards and Poisons Division, 2003).

Location	Year	Duration	Approx. no. of tyres	
Winchester, USA	1983	9 month	6-9 million	
Powys, Wales	1989	14 years	10 million	
Hagersville, Canada	1990	17 days	8 million	

Currently-applied methods of disposal of tyres do not suffice to guarantee environmental safety, hence the prohibition on the disposal of tyres at conventional landfills. In Poland, provisions of the Act of 11 May 2001 (Journal of Laws, 2001) in force in this regard require that tyre manufacturers recover 75% of the tonnage of manufactured tyres coming off the market, with at least 15% of that taking the form of recycling. As of now, the management of used car tyres in Poland entails material recycling (high energy granulation) and energy recovery (burning tyres in cement kilns). It will be clear from this that used tyres are among types of waste capable of exerting the most-severe environmental impact.

A background to that is the presence in tyres of dozens of different synthetic chemicals intended to confer the right properties, affect the flexibility, strength and durability of the product, and facilitate its processing. Table 2 shows the typical composition of the rubber from which tyres are made. The features determining the performance of a tyre, i.e., its resistance to mechanical damage and road conditions (e.g., as regards water and temperature), are also responsible for the difficulties associated with its redevelopment after use (PAS: 107, 2012; Duda et al., 2016).

More specifically, a significant part of the environmental problem posed by rubber is down to additives, which include vulcanising agents, vulcanisation accelerators, vulcanisation accelerator activators, fillers that confer specific mechanical properties or reduce costs of production, softeners that facilitate processing and improve elasticity at low temperatures, anti-aging substances and substances protecting against fatigue. Rainfall provides for the leaching of these substances out of tyres, which thus constitute a significant source of pollutants exerting a negative impact on the environment. According to Wagner et al. (2018), tires may contain additional substances in amount of 5 to 10 mass %. Preservatives (halogenated cycloalkanes), antioxidants (amines, phenols), desiccants (calcium oxides), plasticizers (aromatic and aliphatic esters), processing aids (mineral oils, peptizers) can be leached from the matrix under favorable conditions.

Polycyclic aromatic hydrocarbons (PAHs) are among the substances washed out of tyres both new and used. Literature data suggested that, in circumstances of a basic pH, the concentrations of polycyclic aromatic hydrocarbons leached from rubber chips may exceed limit values set for drinking water. PAHs can derive from soot, special fillers or residual oil. New car tyres differ from used ones in generating slightly higher concentrations of eluted PAHs (1.2 ppb in new tyres and 0.6 ppb in old ones) (Miller & Chadik, 1993; Water Opcert School, 2020; United States Environmental Protection Agency, 2020). In addition, research by Capolupo et al. (2020) conducted on the elution of various components from car tires in the form of microplastics (grain diameter <1 mm) into freshwater and marine leachates confirm the presence of such substances as benzothiazole, 2(3H)-benzothiazolone and phthalimide. Benzothiazole was found in the highest concentrations in the freshwater and marine leachates (2,313 and 1,460 g/L, respectively). In turn, Hennebert et al. (2014) confirms in its research the possibility of leaching phthalic acid esters (PAE) and polycyclic aromatic hydrocarbons from car tires.

Table 2 Indicative composition of a tyre (PAS: 107, 2012).

Ingredient	Passenger car tyre	Lorry/truck tyre	OTR (off-the-road)
tyre	
	[%]	
Rubber / Elastomers (1)	47	45	47	
Carbon black (2)	21,5	22	22	
Metal	16,5	25	12	
Textile	5,5	–	10	
Zinc oxide	1	2	2	
Sulphur	1	1	1	
Additives (3)	7,5	5	6	
Carbon-based materials (4)	74	67	76	
Notes.

(1) Lorry and OTR tyres contain higher proportions of natural rubber than passenger car tyres.

(2) Silica replaces part of the carbon black in certain types of tyres.

(3) Some of the additives include clays, which may be replaced in part in some tyres with recycled rubber crumb from waste tyres.

(4) These approximate totals would be slightly higher if clays were replaced by recycled crumb rubber from waste tyres.

Another group of substances potentially leaching from tyres includes organic compounds containing nitrogen and sulphur in their structure, serving as vulcanising agents, antioxidants and/or gases. Volatile organic compounds are used at the stages during which tyres are manufactured, in order to ensure adequate flexibility and viscosity. However, most VOCs are removed from the tyre as vulcanisation progresses, such that only about 8% of the original total ultimately remains in a tyre (Miller & Chadik, 1993). According to Miller & Chadik (1993) it is possible for there to be leaching from shreds of rubber chemical compounds, such as the aromatics (e.g., ketones) contained in gasoline, as well as carboxylic acids and aniline. According to Gasteiger (2010), during analysis of the leaching of components from car tires into water, aniline was the main identified leaching agent and occurred at levels almost an order of magnitude higher than other identified components. Leaching aniline from tires is also confirmed by Stier, Horgan & Bonos (2020). Hennebert et al. (2014) indicated that the concentration of aniline can reach 5.73 mg/kg in eluates from the shredded tires.

Leachates are also found to contain several other volatile substances, including benzene and methylbenzene. However, no benzene was found in the field leaching test, though the presence of methylbenzene at low concentrations was confirmed. In addition, trimethylbenzene and ethyltoluene were found to be at considerable concentrations. As mentioned above, these processes occur as tyres are in operation on vehicle wheels. However, the appearance of these pollutants in rainwater does not result in high concentrations, on account of the dilution effect.

Literature data (Grefe, 1989; Edil & Bosscher, 1992; Miller & Chadik, 1993; Humphrey & Katz, 1995; Kim, 1995; Selbes, 2009; Turner & Rice, 2010; Hennebert et al., 2014; Selbes et al., 2015; Redondo-Hasselerharm et al., 2018; Liu et al., 2020; Mohajerani et al., 2020) also confirm the presence of elements such as selenium, arsenic, calcium, iron, zinc and chromium in tyres; while silver and mercury were not found. Grefe (1989) reports that zinc, barium, iron and manganese were present in laboratory-modelled processes of leaching. Chromium has also been detected at low concentrations, while those of iron and manganese have both been found to exceed permissible values for drinking water (Water Opcert School, 2020; United States Environmental Protection Agency, 2020).

Kim (1995) presents the results of leaching from rubber chips, with samples taken and tested (filtered or unfiltered) 790 and 830 days on from the onset of the washing process in columns consisting of a mixture of rubber chips and soil or else soil itself. Analyses were performed for such metals as zinc, barium, arsenic, lead and chromium, but concentrations in samples tested were not seen to exceed levels permissible for drinking water. The highest concentration of lead was recorded with unfiltered leaching carried out on a column consisting of a mixture of rubber chips and soil.

Miller & Chadik (1993) also analysed the effect of pH on the quality and level of leaching of chemical compounds from tyres, finding no correlation between pH value and concentrations of leached metal ions. According to the authors, this may reflect the adsorption of metal ions by rubber chips. Other studies (Liu, Mead & Stacer, 1998; Mohajerani et al., 2020) have shown that metals are leached from rubber materials at the highest concentrations where prevailing conditions are acidic. On the other hand, PAHs and petroleum hydrocarbons are leached from rubber materials in highest concentrations where pH conditions are basic. The Environment Agency Regulatory Position Statement 085 indicates that tire bales should not be used below the water level in Source 1 or 2 Conservation Zones or in very acidic environments (peat bogs etc.) where the soil pH is 5 or less, in order to reduce risk of leaching. However, the latest scientific reports include Mohajerani et al. (2020) still indicate the need for further comprehensive research on the leaching of toxic heavy metals and other substances from used rubber products. Indicates the need to test leachate at various pH values and the ratio of liquids to solids.

At present tyre bales are being successfully used in mainly civil engineering applications, in landfill construction where innovation is less constrained than in other sectors of the construction industry, and in unpaved roads (Winter, Watts & Johnson, 2006; Simm, Winter & Waite, 2008). It is rubber material from tyres recycled in the form of bales that is best suited to civil-engineering applications, especially in transport infrastructure, geotechnics and hydrotechnics.

The Faculty of Civil and Environmental Engineering and Architecture of Rzeszow University of Technology has been joined by executive and design companies within the framework of the ReUse research project aimed at developing and implementing an innovative, cheap and environmentally-friendly building material recycled from waste in the form of tyre bales. Engineering constructions filled with tyre bales are exposed to aggressive rainwater, potentially causing leaching of organic and/or inorganic compounds from the tyres generally considered hazardous to the environment and people.

Intact tyres are less prone to leaching than shredded tyre rubber. The leaching research indicates that the available leaching surface on the tyre material is an important leaching factor, especially for zinc and PAH. Decreasing the available surface by using larger fractions of tyre shreds is favourable in an environmental point of view (Edeskär, 2006; Selbes et al., 2015). Large size tire shreds can be also an economical alternative compared to the small size tire shreds in the construction of the tire shred embankment (Khan & Shalaby, 2002). In addidion, Birkholz, Belton & Guidotti (2003) showed that fresh shredded tyre produced a moderate toxic threat to aquatic species, if run-off was not diluted. However, the danger receded as the material aged. However, according to Gualtieri et al. (2005) the growth of tire particles used in the sample does not correlate directly with Zn leaching, because particle aggregation may occur, thus limiting the surface area of exposed granular material. Laboratory tests of heavy metal leaching from new and used tires were also conducted by Fenner & Clarke (2003). Used tires have been exposed to the aquatic environment for thirteen years. The results show that the factor limiting their use may be the release of cadmium and leaching of vulcanizing chemicals. It has been found that tires can be used sustainably in a river or coastal environment, provided that local conditions at the construction site are carefully considered. The purpose of the research carried out and described in this article has been to determine whether tyre bales cause pollution of rainwater, and then groundwater and soil; as well as what level of leaching of undesirable substances takes place.

Materials & Methods

Tyre bales

The pressing and packing of tyres has represented a response to the obligation that the input of tyre-related components be reduced, also given the increasing risk that the tyres present at dumps might self-ignite. Tyre bales considerable potential for use in construction, particularly where their low density and ease of handling give them an advantage. Tyre bales used in the experiment comprised 135 whole waste passenger-vehicle tyres compressed into a lightweight block with the weight of around 1,000 kg and the density of circa 500 kg/m3. Each bale contained only one type and size of tyres. The bales were produced according to PAS 108 (PAS: 108, 2007) in a tyre baler capable of compressing whole tyres to reach the density not lower than 420 kg/m3. The bales measured approximately 1.30 ×2.05 ×0.75 m and were secured by six galvanized steel tie wires running around the length and depth of the bale (Fig. 1). Apart from tyre bale dimensions -slightly modified to facilitate transport - the remaining PAS 108 (PAS: 108, 2007) production requirements, i.e., compressing force, density (mass/volume ratio) and number and tension of steel tie wires, were fulfilled.

Figure 1 Tyre bales used in research.

(A) General view of tyre bale. (B) Detail of connection wires.

The impact of tyre bales on water quality

The environmental testing of tyre bales proceeded in stages. Stage I saw the tyres from tyre bales washed with tap water (WW) (Fig. 2), with both the collected leachate (M) and inputting water made the subject of physico-chemical analyses. The used tyres involved were obtained from a tyre store and not cleaned preliminarily in any way. In Stage II of the testing, washed tyre bales were placed in two separate pools (Fig. 3), i.e., one filled with tap water at pH 8.71 and a second acidified to pH 3.87. The goal was for the first pool to mirror conditions in which rain and surface/ground water is present, while the second simulated situations of acid rain.

Figure 2 Stage I: washing tyres intended to perform tyre bales; testing leachate.

Figure 3 Leachate test trial.

(A) Tyre bale in test pool. (B) Test trial protected against atmospheric influences.

The tyre bales were left immersed for 120 days, with sampling on days 0, 21, 68, 99 and 120. Based on the reviewed literature and previous experience in site contamination, factors that may affect the rate of leaching and/or the concentration of tyre leachate compounds in soil, surface water and groundwater also include contact time with water. The longer the tyres are in contact with water, the greater the risk of groundwater contamination. In our research, the experiment lasted up to 120 days due to the durability of the tires. Also, the half-life times of plastics can reach several dozen days, for example in marine water—60 days, in soil -120 days, in fresh or estuarine water—40 days, in marine sediment—180 days, in fresh or estuarine sediment—120 days (European Commission, REACH Online, 2007; Muniyasamy et al., 2019). The purpose of this experiment was to make sure that the effect of prolonged contact of tires with water would not be leaching harmful components from the tires. Due to the fact that no significant changes in water composition were observed after 120 days, it was decided not to continue the experiment any longer. All the experiments were done in duplicated, with an observed deviation of less than 5%. Additional effluent, formed in the sludge-well located at the bottom of the test setup (Fig. 4)—as intended for mechanical tests of backfill pressure distribution—was also analysed (Duda & Siwowski, 2020a; Duda & Siwowski, 2020b). Samples there were taken after the first flushing of tyres, and on the last day of the trial.

Figure 4 Full-scale trial to evaluate pressure distribution of bridge abutment backfill made of tyre bales.

(A) Installation of the moving wall between measuring chamber and backfill chamber. (B) Filling the backfill chamber with tyre bales. (C) Sampling of effluent water from the well from measuring chamber. (D) Scheme of full-scale trial.

The scope of the physico-chemical analyses of generated waters and wastewaters reflected known technological parameters of tyres, information in the literature, and indicators listed in the EU’s Water Framework Directive. Results were in particular set against provisions in the Polish law implementing the Directive, as well as Directive 2010/75/EU of the European Parliament and of the Council of November 24, 2010 on industrial emissions (integrated pollution prevention and control), as recast.

Where the Polish domestic law was concerned, reference was made to “Regulation 1”, i.e., the Regulation of the Minister of the Maritime Economy and Inland Navigation of 12 July 2019 on substances particularly harmful to the aquatic environment and conditions to be met when introducing sewage into waters or into the ground, as well as when discharging rainwater or snowmelt into waters or for water equipment (the Official Journal of Laws of 2019, item 1311); as well as “Regulation 2”, i.e., the Regulation of the Minister of the Maritime Economy and Inland Navigation of 29 August 2019 on the requirements to be met by surface waters used to supply the population with water intended for human consumption (the Official Journal of Laws of 2019, item 1747)

The comparison of the results obtained with the standards arising from “Regulation 1” is of decisive importance where any further treatment of effluent arising is concerned, while the comparison with standards laid down in “Regulation 2” represents auxiliary information in this regard. The scope and characteristics of methodologies involved in the physico-chemical analysis are as summarised in Table 3.

Table 3 The scope of research and research methodologies used.

Σ16PAHs: naphthalene (Na), acenaphthylene (Acy), acenaphthene (Ace), fluoren (Flu), phenanthrene (Fen), anthracene (An), fluoranthene (Fl), pyrene (Pir), benzo [a] anthracene (BaA), chrysene (Ch), benzo [b] fluoranthene (BbF), benzo [k] fluoranthene (BkF), benzo [a] pyrene (BaP), indeno [1,2,3-cd] pyrene (IP), dibenzo [ah] anthracene (DBA), benzo [ghi] perylene (BghiP).

Parameter	Unit	Method	Method details	
Reaction	pH	In situ meter	pH-meter, MultiLine P4	
Temperature	°C	In situ meter	HQ30D Digital single channel multimeter	
Dissolved Oxygen	mg L−1	In situ meter	HQ30D Digital single channel multimeter	
Conductivity in 20° C	µS cm−1	In situ meter	Conductivity meter, Elmetron IP 67	
Total suspension	mg L−1	PN-EN-872:2007	Samples volume: 100 mL	
Solutes	mg L−1	PN-78/C-04541	Samples volume: 100 mL	
Total Organic Carbon	mg L−1	PN-EN 1484:1999	Samples volume: 20 mL
TOC/N analyzer	
Total Nitrogen	mg L−1	PN-EN ISO 11905-1:2001	Samples volume: 20 mL
TOC/N analyzer	
NH4+, NO3−, Cl−, PO43−, SO43−, Na+, K+, Mg2+, Ca2+,	mg L−1	PN-EN ISO 14911:2002	Samples volume: 100 mL
Ion chromatography analysis	
Dibutyl phthalate, Bis(2-ethylhexyl) phtalate	mg L−1	PN-EN ISO 18856	Samples volume: 100 mL
SPE extraction
GC/MS chromatography analysis	
Σ16PAH’s	mg L−1	PN-ISO 18287	Samples volume: 100 mL
SPE extraction
GC/MS chromatography analysis	
Cd, Zn, Pb, Cu	mg L−1	PN-EN ISO 11885	Samples volume: 100 mL
Mineralization in MARS 6 microwave mineralizer
ICP-OES analysis	

Results

Stage I. Tyre-washing

The test results presented in Table 4 confirm a documentable change in the composition of the washing medium, whose pH after flushing increased from 8.98 to slightly above 9. This value slightly exceeds the permissible recommended limit value for surface water used to supply the population with drinking water and for sewage discharged into waters or into the ground. However, as the water used in washing was already alkaline, given the pH of 8.98, the actual enrichment in alkalizing substances that was involved was limited.

Table 4 Water quality and leachate after washing tyres.

Parameter	Unit	Results	Reg. 1	Regulation 2	
		WW	M		A1	A2	A3	
Reaction	pH	8.98	9.16	6.5–9	6.5–8.5	5.5–9	5.5–9	
Temperature	°C	20.80	25.20	35	25	25	25	
Dissolved Oxygen	% of sat.	7.41	7.53	–	>70	>50	>30	
Conductivity in 20 ° C	µS cm−1	454.00	503.00	–	1000	1000	1,000	
Total suspension	mg L−1	30	115	35	25	30	35	
Solutes	mg L−1	270	345	–	–	–	–	
Total Organic Carbon	mg L−1	0.098	40.9	30	5	10	15	
Total Nitrogen	mg L−1	0.710	5.046	30	1	2	3	
NH4+	mg L−1	0.028	0.365	12.8	0.5	1.5	2	
NO3−	mg L−1	2.717	0.057	102	50	50	50	
Cl−	mg L−1	13.81	21.22	1000	250	250	250	
PO43−	mg L−1	0.828	1.316	9.2 (Pog)	0.4	0.7	0.7	
SO43−	mg L−1	<LOQ	<LOQ	500	250	250	250	
Na+	mg L−1	10.70	13.63	800	–	–	–	
K+	mg L−1	3.24	5.27	80	–	–	–	
Mg2+	mg L−1	12.49	12.23	–	–	–	–	
Ca2+	mg L−1	46.60	53.85	–	–	–	–	
Dibutyl phthalate	mg L−1	<LOQ	<LOQ	–	–	–	–	
Bis(2-ethylhexyl) phtalate	mg L−1	<LOQ	<LOQ	–	–	–	–	
Σ16PAH’s	mg L−1	<LOQ	<LOQ	–	0.0002	0.0002	0.001	
Cd	mg L−1	<LOQ	<LOQ	0.4	0.005	0.005	0.005	
Zn	mg L−1	<LOQ	0.4797	2	3	5	5	
Pb	mg L−1	<LOQ	0.0235	0.5	0.05	0.05	0.05	
Cu	mg L−1	<LOQ	0	0.5	0.05	0.05	0.5	
Notes.

WW “zero” tap water

M leachate from washing tyres from the existing tyre bale (130 pcs.)

+ Worn out individual tyres (130 pcs.)

< < LOQ below the limit of quantification

Increases in electrolytic conductivity, suspension and solutes were also observed, denoting that the initial washing of tyres sees solids and dissolved substances transferred from surfaces into the water. Due to the marked attendant increase in total organic carbon (TOC) from 0.0985 to 40.90 mgC L−1 (M), this leachate does not meet the standards set out in the Regulations cited above. However, the tyre bales were in a dirty state, making it inevitable that large amounts of dirt will be removed by initial washing.

At a more detailed level, wastewater generated was found to have an increased concentration of total nitrogen that exceeded the 5 mg L−1 permissible content for treated waters meeting the needs of the population as stipulated in “Regulation 2”. The content of phosphates in leachate also fails to meet requirements for water serving as a source of water supply. Once again, however, it needs stressing that the determined concentrations were far below those permitted in the case of wastewater.

Furthermore, tyre leachate was not found to contain either PAHs and phthalates used as plasticisers or heavy metals (notably copper and cadmium). Tested concentrations of organics were below the detection levels possible with the method deployed. Meanwhile, lead and zinc were detected, but at concentrations not exceeding threshold values. The presence of the latter metals reflects the composition of tyres, which are 1% by weight of zinc and about 0.005% of lead, on average.

According to Selbes et al. (2015) operation under neutral pH conditions, prewashing of tires, use of larger size tire chips, and removal of metal wires prior to application will reduce the impact of tire recycle and reuse.

Stage II. Analysis of pollutant leaching from tyre bales

Table 5 summarises results for leaching from tyre bales into tap water under slightly alkaline or acid conditions. Tires are the result of vulcanization, which is initiated by temperature. However, during use, this material absorbs an additional amount of energy, which causes that this material becomes harder over time and may crack. In addition to reducing the utility values, cracks release various tire components into the environment. Where the alkaline water was concerned, no exceedances of indicators were noted in relation to “Regulation 1”. There was also no leaching of organic compounds, be these either PAHs or phthalates (as plasticisers). In the research of Gomes et al. (2010) PAH leaching was also negligible. This confirms the fact that the new tires show higher concentrations of organic compounds including PAHs than the old tires (new tires = 1.2 ppb; old tires 0.6 ppb) (Engstrom & Lamb, 1994). On day 21, the TOC concentration increased was elevated—at 16.82 (as opposed to an initial 1.62) mgC L−1. The trend was maintained to 12.57 mg L−1 until day 120 and the trial’s end. These values fall within categories A2 and A3 where water taken for treatment is concerned, meaning a level also present in most Polish rivers.

Table 5 Physicochemical parameters of leachate water: pool 1 and 2.

Parameter		Pool 1	Pool 2	
		Period [days]	Period [days]	
		0	21	68	99	120	0	21	68	99	120	
Reaction	pH	8.71	8.51	8.34	8.27	8.22	3.87	4.41	5.23	6.22	6.25	
Temp.	°C	26.4	25.7	24.30	12.60	–	26.1	26.1	25.60	11.20	–	
DO	mg L−1	6.73	1.28	1.44	1.55	1.98	7.38	2.5	2.81	2.95	4.07	
Conduct.	µS cm−1	430	437	411	343	329	1048	798	702	662	606	
TS	mg L−1	25	10	5	13	3	20	20	2	32	2	
Solutes	mg L−1	29.5	250	238	230	222	405	530	534	441	417	
TOC	mg L−1	1.62	16.82	17.01	13.55	12.57	3.82	18.37	14.25	11.66	10.66	
TN	mg L−1	0.769	1.539	1.447	1.333	1.301	0.784	2.445	3.126	2.827	2.795	
NH4+	mg L−1	0.020	0.158	0.162	0.264	0.366	0.056	0.773	2.183	2.463	2.542	
NO3−	mg L−1	2.394	0	0.030	0	0.100	2.780	2.472	0.267	<LOQ	0.105	
Cl−	mg L−1	14.76	18.20	15.97	14.09	13.72	191.4	204.4	201.7	181.1	177.5	
PO43−	mg L−1	6.047	0.621	0.338	0.187	0.082	0.674	3.312	0.418	0.057	0.020	
SO43−	mg L−1	0	0	1.443	1.259	2.293	<LOQ	<LOQ	46.14	41.78	42.44	
Na+	mg L−1	10.84	10.86	10.39	8.56	8.02	11.42	11.98	10.26	10.42	9.65	
K+	mg L−1	3.19	3.29	3.33	2.84	2.74	3.28	3.39	3.14	3.21	3.04	
Mg2+	mg L−1	11	10.67	10.24	8.63	8.08	11.22	12.49	11.36	11.38	10.6859	
Ca2+	mg L−1	52.67	51.79	36.74	32.36	34.14	54.44	61.80	56.51	56.80	53.60	
Dibutyl phthalate	mg L−1	<LOQ	0.08	<LOQ	<LOQ	<LOQ	<LOQ	0.05	<LOQ	<LOQ	<LOQ	
Bis(2-ethylhexyl) phtalate	mg L−1	<LOQ	<LOQ	<LOQ	<LOQ	<LOQ	<LOQ	<LOQ	<LOQ	<LOQ	<LOQ	
Σ16PAH’s	mg L−1	<LOQ	<LOQ	<LOQ	<LOQ	<LOQ	<LOQ	<LOQ	<LOQ	<LOQ	<LOQ	
Cd	mg L−1	<LOQ	0.010	<LOQ	<LOQ	<LOQ	<LOQ	0.082	<LOQ	<LOQ	<LOQ	
Zn	mg L−1	<LOQ	0.0159	<LOQ	<LOQ	<LOQ	0.110	0.139	<LOQ	<LOQ	<LOQ	
Pb	mg L−1	<LOQ	0.0035	<LOQ	<LOQ	<LOQ	0.015	0.022	<LOQ	<LOQ	<LOQ	
Cu	mg L−1	<LOQ	0.0098	<LOQ	<LOQ	<LOQ	<LOQ	0.024	<LOQ	<LOQ	<LOQ	

The concentration of phosphate increased immediately after tyre bales were inundated, though still remaining compliant with the standards set for wastewater in “Regulation 2”. Equally, the phosphate concentration was actually very low on subsequent measuring days, perhaps indicating a one-off, perhaps incidental, phenomenon reflecting the situation of given individual tyres. No enrichment of water in contact with tyres by the analysed anions or cations was reported, and neither was there apparently significant leaching of the heavy metals tested for. Just after 21 days the presence of tested metals in water was observed, however in very low concentrations. Depaolini et al. (2017) also studied the leaching of heavy metals from rubber. The values they measured were on average very low, often below the detection limit.

The acid water tested for the various indicators also failed to note any “Regulation 1” exceedances. An alkalizing effect of the eluted substances was observed, with the result that the pH of the water rose steadily from 3.87 on day 0 to 6.25 on day 120. This is a desirable phenomenon, suggesting that tyre bales coming in contact with acid rain will in some sense treat it. There was also no reported leaching of organics—either PAHs or phthalates. As with the slightly-alkaline waters, a day-21 elevation of TOC as compared with the initial situation was noted—to 18.37 (as compared with 3.82) mgC L−1. This was still the case on day 120, though the value was back down to 10.66 mg L−1 by then. Selbes et al. (2015) when analyzing the leaching of dissolved organic carbon, dissolved nitrogen and selected inorganic components from used tires, found that the components associated with the rubber part of tires (DOC, DN, zinc, calcium, magnesium etc.) showed an initial fast and then slow release. On the other hand, a constant leaching rate of iron and manganese has been observed, which is attributed to metal wires inside the tires.

However, the obtained values once again fall within categories A2 and A3 where waters taken for treatment are concerned, which is again the level to be observed in most Polish rivers. No enrichment by anions (including phosphates) or cations was to be noted thanks to contact between tyres and water. Nor was leaching of heavy metals reported.

On analysis of the water collected from the sedimentation well, it was clear (Table 5) that first flooding of a tyre bale results in water being alkalized to pH 10.64. However, the enrichment of water in more-specific components is limited, with no exceedances of limit values for wastewater involved. The samples taken on day 120 of the experiment were slightly more contaminated, given the deposition of impurities originating in the tyre bales. In particular, total suspension was higher at 9.0 (as opposed to 1.145) mg L−1), while sulphate concentration was at 273 mg L−1 compared with an initial 0, and water hardness (Mg and Ca concentration) was also greater. In each case, however, the increases were to levels remaining below those allowable for wastewater in accordance with “Regulation 1”. The organic compounds tested for were not found to be present, while concentrations of heavy metals remained were close to the minimum quantifiable limit. The reasons for the decrease or increase of the given parameters are the physicochemical changes that occur during the prolongation of the contact of tires with water. The concentrations of some substances increase as the contact time of the tires with water increases. Some of the substance, e.g., dibutyl phthalate, is leached from the tires and then transformed into other substances or completely degraded. others, in turn, are transformed into insoluble ones, which are, for example, sorbed on the walls of tires or solid particles in water. Sometimes the differences in the amount of a given substance result from the accuracy of the devices used for the determination, especially for low-concentration compounds. Selbes et al. (2015) conducted research on leaching of dissolved organic carbon, dissolved nitrogen and selected inorganic components from used tires. Different tire particle sizes were exposed to leaching solutions with a pH in the range 3.0 to 10.0 for 28 days. DOC and DN leaching was found to be greater for smaller tire chippings. However, the leaching of inorganic components was independent of size. In general, basic pH conditions increased the leaching of DOC and DN, while acidic pH conditions led to increased metal concentrations. Elution was minimal around neutral pH values for all monitored parameters. Although the total amounts of leaching substances were different, the observed leaching rates were similar for all tire chip sizes and leaching solutions.

Discussion

In this study, heavily worn tires that could already be free of mobile forms of pollution were used. The results obtained indicate the environmental safety of their use. No evidence has been obtained that the use of compressed tyre bales increases the concentration of substances in the tires. It has not been found that this material increases the occurrence of metals, chlorides and sulfates, which have an impact on secondary (aesthetic) drinking water standards. Particularly important is the lack of exceedances for zinc, which is used in the production of tires in large quantities (Rhodes, Ren & Mays, 2012). This confirms the findings of other researchers who do not obstruct leaching of this metal from waste tyre bales (Simm, Wallis & Collins, 2004). Hylands & Shulman (2003) summarise the results of laboratory and field studies to determine the level of leachates from tyres. The result indicates that for all regulated metals and organics the results for post-consumer tyres are well below regulatory levels. Substances which could potentially leach from post-consumer tyre materials are already present in groundwater in developed area. Studies suggest that leachate levels for the majority of contaminants fall below the allowable regulatory limits and will have negligible impact on the general quality of water in close proximity to tyres. However, the National Water Research Institute in Canada conducted tests to determine the toxicity of new car tires and their impact on aquatic animals. Research reports published by the Institute show that both old and new tires are toxic to the analyzed rainbow trout. According to the authors of the study, the toxicity of tires is not related to the substances collected during their use or storage, but is only the result of production processes. A higher level of toxicity has been reported for used car tires than for unused new tires (Day et al., 1993). However, research conducted by Redondo-Hasselerharm et al. (2018) indicate that the car tire, including the chemicals associated with this material, did not adversely affect the four freshwater benthic invertebrates. According to the authors, the car tire is a low risk for freshwater benthic invertebrates. However, the potential long-term effects caused by the slow release and gradual environmental increase of bioavailable zinc and other substances caused by aging of the rubber particles are not expressed in these experiments and still require further research.

Conclusions

1. The washing of tyre bales and collection and disposal of effluent generated will suffice to protect the natural environment where the resulting product is embedded against contamination.

2. The effluent from tyre-washing is enriched only slightly at the time of contact, to the extent that visual checks on tyres used to form a bale (with non-selection of the most-soiled tyres) may render washing prior to installation unnecessary.

3. There is no evidence that tyre bales embedded in the ground represent a real or potential source of pollution, or pose a threat that water and soil quality in the vicinity will deteriorate.

4. The quality of sedimentation-well sludge is also satisfactory, to the extent that no treatment as per the regulations for industrial effluent are necessitated.

5. The conditions of the simulation may reasonably be deemed extreme, given immersion and thus constant contact between tyres and water for a period of 120 days. In real conditions, tyre bales will be in contact with water for a few hours, after which various changes associated with self-cleaning, concentration and a density gradient will ensue, to the extent any negative impact on the environment is further curbed.

The authors thank D. Sobala for directing a research program and technical staff for building and supporting the full scale trial.

Additional Information and Declarations

Competing Interests

Author Contributions

Data Availability

The authors declare there are no competing interests.

Aleksander Duda, Małgorzata Kida and Sabina Ziembowicz conceived and designed the experiments, performed the experiments, analyzed the data, prepared figures and/or tables, authored or reviewed drafts of the paper, and approved the final draft.

Piotr Koszelnik conceived and designed the experiments, performed the experiments, analyzed the data, authored or reviewed drafts of the paper, and approved the final draft.

The following information was supplied regarding data availability:

All raw data are available in Tables 4 and 5.

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
