# Peer review of "Application of material from used car tyres in geotechnics—an environmental impact analysis"

_PeerJ, doi:10.7717/peerj.9546_

## Round 0.1 · original submission · Minor Revisions

Reviewers have pointed out some minor corrections and some clarifications. Authors should correct and clarify them in the revised manuscript and submit.

Reviewer 1 ·

Basic reporting

1. Manuscript presents results about important issues of environmental impact of recycled tyres, in particular the use of tyre bales.
2. Paper was written well with good English. Design of the paper is easy to follow.
3. Figures are in good quality, however it is hard to see what figure 4b) presents.
4. Tables are clear and complete.
5. References should be improved with the recent one.

Experimental design

1. This research is consistent with the aims and scope of the journal. It contains environmental issues connected with tyre recycling.
2. The research goal was clearly stated at the end of the introduction.
3. Authors presented wide literature overview, however the part connected with leaching of different substances (lines 95-147) was written on the basis of old references. It should be improved with recent achievements in this field.
4. In general introduction is written well, but the research gap filled should be underlined. I recommend to expand the introduction with information about using the tyre bales. Was there any previous publication about use of tyre bales and their environmental impact?
5. Materials and methods section describe experiment in detail, however some explanation and additional comments are needed:
a) What tyres was used in the experiment? Was only one type of tyres used or mixed types? Were they from passengers car or from trucks? Which sizes where they? Depending on the tyre type there will be some differences in the materials used in the design of the tyre, at this can affect the substances leaching and may be important in the context of the results.
b)It will be good to shortly characterize the methods used in analyzes of environmental impact. They were presented in the table 3, but short description of the methods used, in the context of repetition and sample size, will be useful.
6. The results section presents interesting results describing environmental impact of tyres bales in the context of leaching. Authors clearly present the complete results. The are, however, some issues need to be improved:
a) the discussion and comparison of the results with previous study should be done,
b) it is not clear now if the time of staying bales in the water is important for the leaching phenomenon. Why 120 days of experiment was set?
c) the reasons of the increasing or decreasing levels of substances can be described.

Validity of the findings

a) All underlying data have been provided; they are robust, statistically sound, & controlled.
b) Conclusions are well stated, linked to original research question & limited to supporting results, however it can be improved according to my comment 6 from Experimental design.

Annotated reviews are not available for download in order to protect the identity of reviewers who chose to remain anonymous.

Reviewer 2 ·

Basic reporting

clear

Experimental design

within Aims and Scope of the journal

Validity of the findings

innovative

Additional comments

In this paper, the environmental impact analysis from used car tyres in geotechnics was conducted. This research is innovative. But some issues should be solved:

Introduction: Authors should remove unnecessary information. For more information regarding generation and utilization of on waste rubber and more application of rubber tire, you should refer and cite the following paper:

Liu, L., Cai, G., Zhang, J., Liu, X., & Liu, K. (2020). Evaluation of engineering properties and environmental effect of recycled waste tire-sand/soil in geotechnical engineering: A compressive review. Renewable and Sustainable Energy Reviews, 126, 109831.

Liu, L., Cai, G., & Liu, X. (2020). Investigation of thermal conductivity and prediction model of recycled tire rubber-sand mixtures as lightweight backfill. Construction and Building Materials, 248, 118657.

Liu, L., Cai, G., & Liu, S. (2018). Compression properties and micro-mechanisms of rubber-sand particle mixtures considering grain breakage. Construction and Building Materials, 187, 1061-1072.

Tyre bales: Description for the tyre bales is not clear. Please add more information.

Results and Discussion: Authors were suggested to add the approximate reason for the results.

Reviewer 3 ·

Basic reporting

Application of material from used car tyres in geotechnics - environmental impact analysis

the paper met the standards of the journal
even though the writing is ambiguous confusing in some parts
the the results shows interesting alternate material

Experimental design

the experimental method is adequate
and investigation performed to support the objective

Validity of the findings

the findings are good and recycle the waste tyres is though not novel its impact on environment is being analysed

Additional comments

The purpose of the research carried out and described in this article has been to determine whether tyre bales cause pollution of rainwater, and then groundwater and soil; as well as what level of leaching of undesirable substances


This should be highlighted in the abstract itself


149 paragraph is repeated
172 Tyre bales are potentially a new building material from the recycling market in transport- confusing
modify
apart from that ensure that the format of the journal is maintained

---

## Round 0.2 · accepted · Accept

Authors have carried out revision as per comments of reviewer and editor.